# Investigating mental defeat in individuals with chronic pain: Protocol for a longitudinal experience sampling study

Jenna L Gillett ,[1] Paige Karadag ,[1] Kristy Themelis ,[1] Yu-Mei Li ,[2] Sakari Lemola ,[2] Shyam Balasubramanian ,[3] Swaran Preet Singh ,[4] Nicole K Y Tang [1]

JLG and PK contributed equally.

JLG and PK are joint first authors.

¹Department of Psychology, University of Warwick, Coventry, UK
²Department of Psychology, Bielefeld University, Bielefeld, Germany
³University Hospitals Coventry and Warwickshire NHS Trust, Coventry, UK
⁴Mental Health and Wellbeing, University of Warwick, Warwick Medical School, Coventry, UK

**Correspondence to**
Prof Nicole K Y Tang;
N.Tang@warwick.ac.uk

## ABSTRACT

**Introduction** Previous qualitative and cross-sectional research has identified a strong sense of mental defeat in people with chronic pain who also experience the greatest levels of distress and disability. This study will adopt a longitudinal experience sampling design to examine the within-person link between the sense of mental defeat and distress and disability associated with chronic pain.

**Methods and analysis** We aim to recruit 198 participants (aged 18–65 years) with chronic pain, to complete two waves of experience sampling over 1 week, 6 months apart (time 1 and time 2). During each wave of experience sampling, the participants are asked to complete three short online surveys per day, to provide in-the-moment ratings of mental defeat, pain, medication usage, physical and social activity, stress, mood, self-compassion, and attention using visual analogue scales. Sleep and physical activity will be measured using a daily diary as well as with wrist actigraphy worn continuously by participants throughout each wave. Linear mixed models and Gaussian graphical models will be fit to the data to: (1) examine the within-person, day-to-day association of mental defeat with outcomes (ie, pain, physical/social activity, medication use and sleep), (2) examine the dynamic temporal and contemporaneous networks of mental defeat with all outcomes and the hypothesised mechanisms of outcomes (ie, perceived stress, mood, attention and self-compassion).

**Ethics and dissemination** The current protocol has been approved by the Health Research Authority and West Midlands—Solihull Research Ethics Committee (Reference Number: 17/WM0053). The study is being conducted in adherence with the Declaration of Helsinki, Warwick Standard Operating Procedures and applicable UK legislation.

## STRENGTHS AND LIMITATIONS OF THIS STUDY

⇒ This study provides the first longitudinal investigation of mental defeat in chronic pain to shed light on its temporal links with outcomes.
⇒ A range of outcomes and hypothesised mechanisms will be assessed including pain, physical and social activity, medication use, sleep, stress, mood, attention and self-compassion. Measures are repeated over a 1-week period, at two time points each 6 months apart.
⇒ This study will use both self-reported and objective estimates of sleep and physical activity, via diaries and actigraphy longitudinally.
⇒ The research is done remotely, at the participant's convenience and within their natural environment.
⇒ Considerations must be given to effects of participants' COVID-19 exposure on recruitment, subsequent attrition and possible findings despite having had appropriate COVID-19 screening and health and safety procedures in place.

## INTRODUCTION

Chronic pain is characterised as pain that persists or recurs beyond 3 months.[1] It is highly prevalent, affecting around 30% of the population worldwide.[2–4] Chronic pain conditions, namely low back pain and headaches, are consistently the top causes of years lived with disability[2 5] and reduced quality of life.[6] People with chronic pain are three times as likely to have depression and anxiety disorders[7] and two times as likely to present a risk of suicide compared with the general population.[8] While some individuals manage to cope with the pain, others struggle to maintain daily activities. Understanding the factors that determine whether an individual can feel and function well—despite persistent pain—is crucial to advancing non-pharmacological management approaches for chronic pain, which have so far had a modest impact.[9]

A concept proposed to help explain differences in the experience of pain-related distress and disability is mental defeat; a cognitive construct characterised by negative self-appraisals in relation to pain.[10 11] The concept of defeat has its theoretical underpinnings in the study of post-traumatic stress disorder

(PTSD)[12–14] and depression,[15 16] where it is respectively defined as the perceived loss of autonomy and a natural response to the loss of social status in a conflict situation. Empirical research has shown that a strong sense of mental defeat is associated with severe PTSD symptoms and poorer response to exposure treatment.[12 14 17] The perception of defeat has also been shown to predict symptoms of depression independent of hopelessness,[15] and has been implicated in psychological models of suicidal behaviour and suicidality.[18–20]

Mental defeat in the context of chronic pain encapsulates people's psychological response to perceived threats of one's physical and psychological autonomy. Daily experience of living with persistent and debilitating pain, which does not respond to treatment, is thought to be a repeated trigger of mental defeat, prompting negative appraisals of self in relation to pain.[10] Qualitative explorations of patients' experiences of pain have consistently revealed patients reporting 'defeat of the mind' and 'the pain is taking over', with pain seen as 'an enemy' that 'belittles (them) as a person'.[11 21 22]

Using the Pain Self Perception Scale to measure mental defeat, it has been found that treatment-seeking patients with chronic pain have elevated levels of mental defeat compared with patients with acute pain, patients with anxiety disorders, community volunteers with chronic pain, community volunteers with acute pain and pain-free volunteers.[10] Mental defeat has also been found to be the predictor explaining the most variance in pain interference, depression and psychological disability among chronic pain patients seeking specialist treatment, when compared with pain intensity, health anxiety, worry rumination and pain catastrophising.[11] It has moderate associations with sleep disturbances and functional disability[11] and negatively relates to pain self-efficacy even when anxiety, depression, pain catastrophising and hopelessness are controlled for.[23] Furthermore, mental defeat predicts suicide intent in patients with chronic pain above and beyond pain intensity.[24] In pain-free volunteers, an activated sense of mental defeat appears to operate independently from existing pain-related psychological constructs such as pain catastrophising, in influencing mood and attentional disengagement from nociceptive stimuli.[25] Together, these findings suggest how a person's self-perception in relation to pain matters in terms of predicting and explaining outcomes.

However, most of the afore-mentioned studies of mental defeat in chronic pain are cross-sectional in design, and more direct evidence is required to establish the temporal and casual association implicated. This study will be the first to use experience sampling methodology (ESM) involving in vivo data gathering (using actigraphy, sleep diaries and daily online surveys) to examine the day-to-day association between mental defeat, symptoms, distress and disability associated with chronic pain. Data will be collected over 7 days, allowing the study of any temporal within-person relationships, which may provide insight into clinically relevant questions such as whether

mental defeat will be followed by higher pain, increased stress, greater attention to pain, increased medication usage, reduced physical and social activity and poorer sleep. The experience sampling exercise is repeated at 6 months to investigate how these indices may change over time and translate into distress and disability long term. The primary objectives of this study are: (1) examine the within-person, day-to-day association of mental defeat with outcomes (ie, pain, physical/social activity, medication use and sleep) and (2) examine the dynamic temporal and contemporaneous networks of mental defeat with all outcomes and the hypothesised mechanisms of outcomes (ie, perceived stress, mood, attention and self-compassion).

Based on previous work,[10 11 25] we hypothesise that, for an individual, a strong sense of mental defeat will be associated with subsequent greater reports of pain, reduced physical and social activity, possible increased use of medication and poorer sleep. Examinations of the dynamic temporal and contemporaneous networks of mental defeat with mechanisms and outcomes are novel and exploratory.

## METHOD
### Study design
This study uses a within-study design that uses an ESM approach.[26] As depicted in figure 1, ESM are used to gather data prospectively over 1 week; at two-time points, each 6 months apart (T1 and T2). The length and frequency of assessment reflect our attempt to balance information needs with concerns of participation burden and possible attrition, participants are asked to continuously wear a medical-grade actigraphy device (MotionWatch 8, manufactured by CamNTech) for 8 nights/7 days. They are asked to complete a sleep diary and respond to three short online surveys during the day. These surveys will provide in-the-moment measures of mental defeat, pain, medication use, physical activity and social activity, stress, mood, attention and self-compassion using visual analogue scales (VAS).

Participants are adults aged 18–65 living with chronic non-cancer pain that has been present or recurring for more than 3 months.[27 28] Participant inclusion and exclusion criteria are outlined in box 1.

In terms of sample size calculation, based on running previous ESM studies of this kind,[29] we aim to recruit 198 participants to factor in an anticipated 20% attrition at T2. This will give an estimated 158 participants who will complete the experience sampling procedure at both time points to generate up to 6636 temporally structured survey ratings for analysis (3 ratings×7 days×2 time points×158 participants). There will be a maximum of 2528 observations of sleep data (8 nights of actigraphic data×2 time points×158 participants) and 7 days' of physical activity count data at 30 s epoch. This will give sufficient power to perform the planned analyses using Multilevel Mixed-Effects Models[30] and Graphical Gaussian Models.[31]

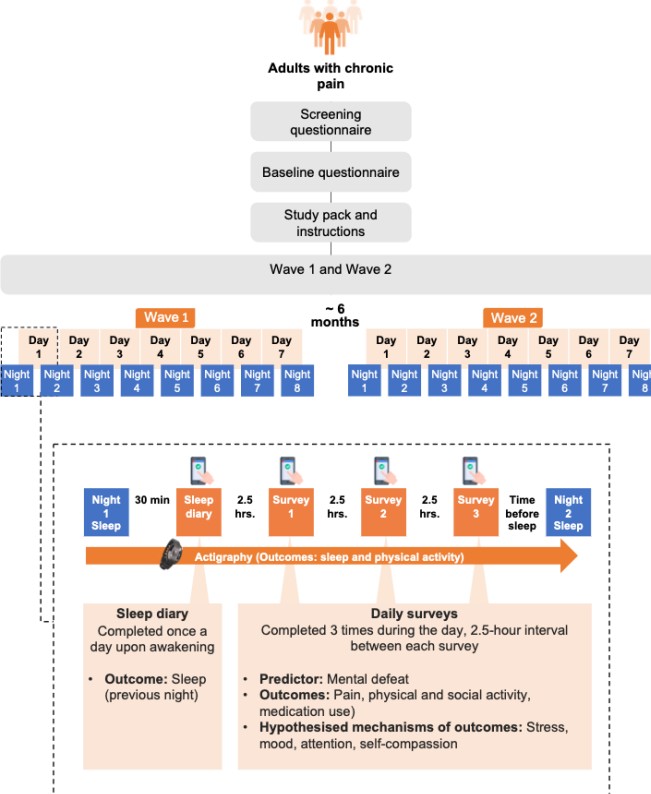

**Figure 1** The study uses a prospective experience sampling design involving in vivo data gathering using survey and actigraphy. The participants are asked to engage with the data collection process over 7 consecutive days (8 nights) two times, 6 months apart. An example data collection procedure in a single day is detailed in the box of dashed outline. The participants are prompted to complete a sleep diary in the morning and three daily surveys each day. The participants are also asked to wear an actigraph during the entire 7-day (8-night) period for each wave, generating objective estimates of sleep and physical activity. The timing of the diary and surveys is prespecified. If a participant's typical wake time is 08:00, a prompt to complete the sleep diary will be sent at 08:30, then the first, second and third daily surveys at 11:00, 13:30 and 16:00, respectively.

## Recruitment

Recruitment of participants started in April 2021 and is expected to end in May 2023. A variety of recruitment methods are being used, including social media, the NIHR Clinical Research Network, public engagement events and peer-led support groups. We are also using online recruitment platforms to capture individuals with chronic pain with registered interest to take part in research. Finally, chronic pain patients at University Hospitals Coventry and Warwickshire are given information about the study during pain clinic appointments.

## Measures

### Screening questionnaire

A brief online screening questionnaire is administered to assess eligibility. To provide the relevant information against the a priori inclusion and exclusion criteria, the screening questionnaire determines basic demographics

---

**Box 1 Participant inclusion and exclusion criteria for participation**

**Inclusion criteria**

⇒ Aged between 18 and 65 (for focussing our study on working-age population).
⇒ Experience chronic non-cancer pain for at least 3 months.
⇒ Stable treatment for duration of the study (6 months).
⇒ English-speaking (for understanding and implementing the data collection procedure).
⇒ Living in the UK (for postage of equipment).
⇒ Be able to provide informed consent.

**Exclusion criteria**

⇒ Have any significant comorbid psychiatric (eg, psychosis), medical (eg, coronary heart diseases), neurological (eg, Alzheimer's, Parkinson's, Epilepsy) or life-threatening conditions that would impact pain experience, impede the ability to provide informed consent or complete the study.
⇒ Have any other significant comorbid sleep disorder, for example, sleep apnoea, restless leg syndrome, periodic limb movement disorder, narcolepsy or circadian rhythm disorders, which in the opinion of the research team would cofound the results of the study.
⇒ Have elective surgery or procedures requiring general anaesthetic during the study.
⇒ Have participated in another research study using an investigational product in the past 3 months.

Note. The examples given in the inclusion/exclusion criteria are not exhaustive and participants' eligibility is assessed on a case-by-case basis by the research team.

---

(eg, age, sex, ethnicity, employment status, education level) and health indicators (eg, body mass index, average alcohol intake, smoking status). It also checks for pain characteristics, current treatment plans, for example, plans for surgery in the next 6 months and current participation in clinical trials. Lastly, it considers comorbid health conditions, including the presence of any psychiatric, medical, neurological or sleep disorders.

### Baseline questionnaire

Eligible participants are asked to complete a baseline questionnaire, which includes validated measures of variables related to mental defeat, pain, physical and social activity, sleep, psychological states and quality of life (see table 1 for full list). Data from these questionnaires are not used in the planned analyses of the current study except for characterising the sample at baseline.

### Sleep diary

The morning section of the Consensus Sleep Diary[32] is administered daily throughout the tracking period to collect self-reported information on sleep. The sleep diary asks participants what time they went to bed, what time they attempted to go to sleep, how long it took them to fall asleep (sleep onset latency), how many times did they wake up from sleep (not including final awakening), final wake up time, what time did they get out of bed, total sleep duration (hours and minutes), perceived

**Table 1** Questionnaire measures included

| Measure | Scale used |
|---|---|
| *Key variable of interest* | |
| Mental defeat | Pain Self-Perception Scale[10] |
| *Pain-related measures* | |
| Pain intensity and interference | Brief Pain Inventory-Short Form[47] |
| Pain vigilance and awareness | Pain Vigilance & Awareness Questionnaire[48] |
| Pain-related fear of movement | Tampa Scale of Kinesiophobia-11[49] |
| Patterns of activity (pain specific) | Patterns of Activity Measure for Pain[50] |
| Pain catastrophising | Pain Catastrophizing Scale[51] |
| Pain self-efficacy | Pain Self-Efficacy Questionnaire[52] |
| *Physical and social activity measures* | |
| Physical activity | International Physical Activity Questionnaire[53] |
| Social activity | Social Activity Log[54] |
| *Sleep-related measure* | |
| Insomnia symptom severity | Insomnia Severity Index[55] |
| *Psychological states* | |
| Stress | Perceived Stress Scale[56] |
| Anxiety and depression | Hospital Anxiety & Depression Scale[57] |
| Suicidal behaviour | Suicidal Behaviour Questionnaire Revised[58] |
| Self-compassion | Self-Compassion Scale Short Form[59] |
| *Quality of life measure* | |
| General health and quality of life | EQ-5D-5L[60] |

sleep-quality and how rested or refreshed they felt after waking. Participants can also provide additional information that they feel is relevant to their sleep. We added two extra questions to the sleep diary to obtain in-the-moment ratings of pain and mood on waking. These possible covariates are assessed via two VAS both ranging from 0 to 10, whereby for pain 0=no pain at all and 10=worst pain imaginable and for mood 0=very bad and 10=very good.

### Daily survey

The daily survey allows participants to provide self-report ratings at multiple points throughout each day. We use adapted or proxy measures as well as shortened versions of original scales to decrease participant burden and determine momentary assessments of mental defeat, pain, medication use, physical and social activity, stress, mood, attention and self-compassion (see figure 1 for a schedule of administration). As part of the PPI piloting process, the selected questions and proxy measures were approved by our PPI representatives (two people with lived experience of chronic pain). The surveys are short (<5 min completion time) and are equally spaced 2.5 hours apart to capture experiences at different time points throughout the day. The surveys are sent out following a choice of predetermined schedules to match participants' typical sleep–wake patterns. The earliest schedule starts with a sleep diary at 06:30 and the first daily survey commences at 09:00, whereas the latest schedule starts with a sleep

diary at 13:30 and the first daily survey commences at 16:00. The timing of these measures avoids unsociable hours, as no prompts arrive between the hours of 23:00 and 06:00 inclusive.

We use survey signal to send out autoprompts at specified times via SMS to participants' smartphones. This enables accessibility for participants to complete surveys in a timely fashion, while remaining convenient, and does not require an app download or adjusted personal mobile settings. The surveys are administered, recorded and returned via Qualtrics and are time stamped at the time of commencement and completion. The daily surveys comprise multiple VAS with varying left to right anchors, as shown in table 2.

### Actigraphy

Actigraphs are light, compact accelerometer-based devices that have been used to generate objective estimates of sleep parameters for several decades.[33] Actigraphy has been well-evidenced as a suitable methodology for non-intrusive at-home sleep assessment.[34–36] Although polysomnography continues to be considered the gold standard for sleep recording, wrist actigraphy has the advantage of offering cost-effective continuous recording in participants' home environment.[37] Thus providing more ecologically valid information compared with polysomnography.[37]

**Table 2** Daily survey self-report rating scales

| Construct | Item (measure) | Scale | Anchors |
|---|---|---|---|
| Mental defeat | Since waking up today, how much has the pain brought back to life memories of times when you felt the pain had taken over? | 0–10 | 0=not at all to 10=very much so |
| Pain intensity | What is your current pain level? | 0–10 | 0=no pain to 10=worst pain imaginable |
| Pain interference | Since waking up today, how much has your pain impacted on…<br>1. Your daily routine (including work)?<br>2. Your relationship(s)?<br>3. How you think or feel about the future?<br>4. How you think or feel about yourself? | 0–10 | 0=not much impact/interference to 10=a great deal of impact/interference |
| Medication use | Since waking up today, would you say that you have taken more or less medication than usual? | −5-5 | −5=a lot less than usual, 0=no difference, 5=a lot more than usual |
| Physical activity | Since waking up today, how physically active have you been? | 0–10 | 0=not physically active at all to 10=very physically active |
| Social activity | Since waking up today, how socially engaged have you been? | 0–10 | 0=not socially engaged at all to 10=very socially engaged |
| Stress | What is your current stress level? | 0–10 | 0=no stress at all to 10=highest level of stress possible |
| Mood | What is your current mood? | 0–10 | 0=very bad to 10=very good |
| Attention to pain | Since waking up today, how much of the time have you been thinking about your pain? | 0–10 | 0=none of the time to 10=a great deal of the time |
| Focus of attention | Since waking up today, has the focus of your attention been…<br>1. Inward or outward<br>2. On the body or mind | 0–10 | 0=inward to 10 outward<br>0=body to 10=mind |
| Self-compassion | Since waking up today, how kind…<br>1. To yourself have you been?<br>2. To others have you been? | 0–10 | 0=not at all to 10=very much so |

Note. VAS questions with corresponding anchors that are presented in survey 1 are summarised above. These questions appear identically in surveys 2 and 3, except instead of starting questions with 'since waking up today…' in surveys 2 and 3 each question begins 'in the last 2.5 hours…'

For the present study, MotionWatch 8 actigraphs are worn by participants on their non-dominant wrist during the study. The MotionWatch is a medical-grade triaxial actigraphy device containing a piezoelectric accelerometer to record duration, integration and number of movements in all directions. This data enables the research team to chart sleep and physical activity across the experience sampling periods that are then downloaded for analysis using MotionWare software (Cambridge Neurotechnology, Cambridge, UK) with validated algorithms. The key sleep parameters we are interested in are sleep efficiency and total sleep time. The key physical activity parameter is total activity counts tabulated by week, day and/or hour.

### Procedure

To participate, individuals respond to a study advert via phone, email or by following a direct link to the information leaflet. Interested participants are required to complete the screening questionnaire and contact information form, following which a member of the research team determines eligibility to the study by checking against the inclusion/exclusion criteria. Individuals who meet the inclusion criteria are informed of their eligibility via email and invited to participate. Individuals who do not meet the inclusion criteria are informed via email that they are not eligible, thanked and debriefed.

Eligible participants that agree to participate are sent a link to an online consent form to complete via Qualtrics. Once informed consent has been obtained, participants have a phone call with a member of the research team to arrange their participation and highlight some key training points for the study. All participants receive the study materials (an invitation letter, a cleaned and packaged MotionWatch for borrowed use, two individually wrapped disinfectant wipes, an addressed, prepaid return envelope and an information booklet) via UK postal delivery. To accompany the information booklet, participants are emailed a link to an instructional video

(see online supplemental file 1) demonstrating how to use the MotionWatch devices appropriately and reiterating the schedule of measurement/engagement required during participation. Participants are instructed to wear the MotionWatch continuously for the 7-day experience sampling period and are required to press an event marker on the device when they plan to go to sleep and when they get out of bed, following their main sleep period. Participants are also asked to answer the sleep diary and three short surveys each day.

The timing of the daily surveys/sleep diary are individually anchored by participants' typical sleep–wake patterns, to accommodate and control for variations in individuals' circadian rhythms. For example, if a participant indicates that they usually wake at 08:00, a typical day during the experience sampling period on this schedule would be as follows: on getting out of bed, the participant presses the event marker on the MotionWatch and will receive the first text message (with a link to the sleep diary) at 08:30—allowing them to report their sleep experiences as soon as practical after waking. Throughout the day, the participant will receive three further text message prompts, each containing a link to the short online survey. Survey 1 (S1) is received at 11:00, survey 2 (S2) at 13:30 and survey 3 (S3) at 15:00. The links to each survey remains open for 2.5 hours, before expiring at the time the following survey is triggered. Finally, the participant will be required to press the event marker at 'lights out' or when beginning trying to sleep. This process is identical for each day in the experience sampling period. After the final awakening on the last day, the participant removes the MotionWatch, and packages it in the box ready for return postage to the Lab. The MotionWatch data is processed on-site at the Lab, before being formatted into a personalised breakdown for the participant, which is emailed to them along with a gift voucher within 2 weeks of receipt of the returned equipment. This procedure is repeated in its entirety for the follow-up T2 assessment. The debrief is administered on study completion.

### Participant reimbursement

To thank the participants for their time and participation, they are given a £10 gift voucher for each time point they complete. Additionally, participants are provided with a personalised breakdown of their actigraphy data created by the research team. No evaluative feedback on sleep quality and physical activity patterns is given.

### Adverse event recording and management

This is a low-risk observational study, and no major adverse events are anticipated. We offer health and safety training at the outset, instructing participants not to respond to survey text messages if it is not safe to do so, for example, when driving, operating machinery or crossing the road. Before participation commences, participants are instructed to report any adverse events that occur during the assessment periods to the research team.

Adverse events that are related to the study and/or unrelated adverse events are recorded and reported to the study sponsor according to reporting requirements. Unrelated and expected adverse events may include but are not limited to illness, hospitalisation or day surgery that occurs during the assessment period. An adverse event that a small number of participants may experience is discomfort or irritation caused by wearing the Motion-Watch. Before participants begin the study, we advise them to inform the researcher if they experience any skin irritation. If the irritation is very mild and they wish to continue, we recommend placing a small piece of tissue underneath the watch or to place the silicon strap on top of their sleeve to avoid direct skin contact. In the unlikely event of an unexpected adverse event that is deemed to be severe and related to the study, the research team would immediately pause the study and send an expedited report to the study sponsor. Events will be followed up until they are resolved or when a final outcome has been reached.

### Patient and public involvement

This protocol has been developed in partnership with two patient representatives with lived experience of chronic pain. One representative (PR) provided feedback on participant-facing materials and piloted the MotionWatch for at-home use. Debra Dulake participated in study-related procedures and also commented on the manuscript for readability.

### COVID-19-related considerations

To enable the study to take place during the COVID-19 pandemic, while adhering to governmental and institutional COVID-19 guidance, the operational aspects of the protocol have been adapted for remote participation. The MotionWatch devices are all sanitised, prepared and packaged for participation in a lab environment. A short COVID-19 checklist has been implemented before each wave of tracking (see online supplemental file 2).

## DATA MANAGEMENT PLAN

The study data collected from the questionnaire, sleep diary and daily surveys will be stored securely on Qualtrics and subsequently downloaded as password protected databases to undergo data quality checks and pseudo-anonymisation by the research team. Access to these databases will be restricted to approved members of the research team. Once data completeness and quality are verified, electronic data on Qualtrics will be deleted.

Actigraphic data collected using MotionWatch 8 are downloaded using Motionware software on each watch's return to the lab. The downloaded data are saved via their assigned ID number and will be held securely and separately from the study data.

The chief investigator (NKYT) of the project will act as the data controller. All data generated by the research programme will be analysed by the research team either on

site at the University of Warwick or in a private workspace in the event of remote working, as per the University's Off-Campus Working Policy. In line with the University's Research Code of Conduct, data will be retained in electronic format for at least 10 years from the date of any publication that is based on it.

## STATISTICAL METHODS

All participants who meet eligibility criteria will be included in analyses; including those who wish to withdraw from the study but consent to having any data already collected analysed. Those who withdraw and do not consent to the data being analysed will be excluded from analyses.

Descriptive statistics will be used to characterise the sample based on information from the screening and baseline questionnaires. Means and SD/95% CIs will be reported for continuous variables, whereas frequencies and percentages will be used for reporting categorical variables.

To evaluate the within-person temporal relationship between mental defeat (predictor) and pain, physical and social activity, medication use and sleep (outcomes), we will pool the daily survey ratings from all participants across waves of assessment. Linear mixed models with a time-lagged design will be fit to the data. We will fit one model for each outcome. For each analysis, we will first estimate a maximal random effects model with all random slopes and random intercepts (for 'time of day (survey 1, 2, 3)', 'day (1, 2, 3, 4, 5, 6, 7)' and 'wave (1, 2)') and successively simplify the random effect structure until the model converges. Nested comparisons will be made between the final models with the intercepts-only models. The significance of each model will be assessed using a likelihood ratio test. P values will be adjusted using false discovery rate to account for alpha error accumulation[38]. The best fitting model will be determined using Akaike Information Criteria (AIC)[39 40] and Bayesian Information Criteria (BIC),[41] where lower values of the AIC and BIC indicate a better fit. As secondary analyses, we will repeat the above analyses with the hypothesised mechanisms of outcomes (stress, mood, attention, self-compassion) as the dependent variables. Both the statistical software SPSS v.28 and the 'lme4' package[42] for R[43] will be used.

To explore the within-individual temporal and contemporaneous relationships as well as between-individual relationships between mental defeat and the outcomes and hypothesised mechanisms of interest, we will use Gaussian graphical models (GGM[31]). To obtain sufficient statistical power for within-in individual variances in the GGMs, it is recommended to have more than 20 measurements and at least 20 pairs of comparisons per participant.[44] The design of the current study will yield 21 measurements per participant per wave. That will give 20 pairs of comparisons for modelling changes across days, 14 pairs of comparisons for modelling changes within a day and 7 pairs of comparisons for modelling changes at

different times of the day. We will therefore not seek to model changes at different times of the day, but to focus on modelling changes across and within a day by pooling together data from both waves of assessment.

We will use the mlVAR (V.0.3.2) package,[44 45] or similarly suitable but more up-to-date packages, for R[43] to analyse the multivariate time series data. We will report three kinds of network; temporal (to indicate whether a variable predicts another variable (or itself) at the next measurement point, controlling for all other variables in the network at the previous measurement point), contemporaneous (to indicate the within-person relationships between variables, having adjusted for the effect of all variables in the network and the temporal effects) and between-person (to indicate relationships between person-means of variables, partialling out the effect of all other variables in the network).

Additional secondary analyses will be performed for research questions other than the two main ones stated here. Any deviation(s) from the original statistical plan will be described and justified in our subsequent reportings.

## LIMITATIONS

Although this study design provides the first prospective investigation of mental defeat in chronic pain, more research is needed to evidence any potential causality of key mechanisms and outcomes. As in previous experience sampling studies that measure sleep, we expect some minor discrepancies between actigraphy and diary data[46] when self-reporting on one's own sleep estimations, which is a well-documented phenomenon. In this study we will cross-check for any discrepancies between diary and objective sleep measures and will run sensitivity analyses to determine effective interpretation in subsequent analyses. Furthermore, participants are required to undergo some short training on how to participate in the study and use the equipment (actigraphy devices) effectively, but errors and inconsistencies in individual engagement levels are inevitable, so we are expecting some data to be lost to missingness. Finally, considerations must be given to effects of participants' COVID-19 exposure on recruitment, subsequent attrition and possible findings despite having had appropriate COVID-19 screening and health and safety procedures in place.

## ETHICS AND DISSEMINATION

This research forms part of the wider MRC-funded Warwick Study of Mental Defeat in Chronic Pain ('WITHIN' Study). The current protocol has been approved by the Health Research Authority and West Midlands—Solihull Research Ethics Committee (Reference Number: 17/WM0053, p, IRAS project ID: 223190). The University of Warwick (Research Impact Services, University of Warwick, Coventry, CV4 7AL) acts as the Sponsor for the study. The study is being conducted in adherence with the Declaration of Helsinki, Warwick Standard Operating Procedures and applicable UK legislation. Results from this study will be

written as reports, to be disseminated in peer-reviewed journals, at conferences and patient and public engagement events.

**Acknowledgements** The authors would like to thank all the participants who are involved in the study and the UK chronic pain charities who have shared our study on relevant webpages and social media platforms, including Pain Concern the charitable partner of this study. We would also like to thank our PPI representatives (PR and Debra Dulake) for their input into the study design and feasibility, Debra Dulake also commented on the manuscript. Finally, we would like to thank Michelle Pun, Emily Ashton and Chloe Osei-Cobbana who have supported the data collection process under the Psychology Research Skills Development Scheme (PRSDS) at the University of Warwick.

**Contributors** NKYT conceived the research idea and developed the theory and plan for this study. JLG and PK are joint first author as they contributed equally to all aspects of the study and are responsible for implementing the protocol, creating study materials, data acquisition and management and drafting the original manuscript. KT and NKYT are responsible for critical revisions of the manuscript. All authors (JLG, PK, KT, Y-ML, SL, SB, SPS and NKYT) contributed to the study development and reviewed, commented, and approved the manuscript.

**Funding** This protocol is part of a wider study (Warwick Study of Mental Defeat in Chronic Pain that is supported by Medical Research Council (MRC) grant number: MR/S026185/1 to NKYT. For the purpose of open access, the corresponding author has applied a Creative Commons Attribution (CC BY) licence (where permitted by UKRI, 'Open Government Licence' or 'Creative Commons Attribution No-derivatives (CC BY-ND) licence' may be stated instead) to any Author Accepted Manuscript version arising from this submission. JLG was funded by the University of Warwick Departmental PhD Fellowship, under the supervision of NKYT.

**Competing interests** None declared.

**Patient and public involvement** Patients and/or the public were involved in the design, or conduct, or reporting, or dissemination plans of this research. Refer to the Method section for further details.

**Patient consent for publication** Not applicable.

**Provenance and peer review** Not commissioned; externally peer reviewed.

**ORCID iDs**
Jenna L Gillett http://orcid.org/0000-0002-7115-9938
Paige Karadag http://orcid.org/0000-0003-2655-7109
Kristy Themelis http://orcid.org/0000-0002-0022-5272
Yu-Mei Li http://orcid.org/0000-0002-4673-1082
Sakari Lemola http://orcid.org/0000-0002-1314-6194
Shyam Balasubramanian http://orcid.org/0000-0001-5149-3595
Swaran Preet Singh http://orcid.org/0000-0003-3454-2089
Nicole K Y Tang http://orcid.org/0000-0001-7836-9965

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
