## [Reviewer comments · BMJ Open]

ARTICLE DETAILS

TITLE (PROVISIONAL)	Investigating Mental Defeat in Patients with Chronic Pain: Protocol for a Longitudinal Experience Sampling Study
AUTHORS	Gillett, Jenna; Karadag, Paige; Themelis, Kristy; Li, Yu-Mei; Lemola, Sakari; Balasubramanian, Shyam; Singh, Swaran; Tang, Nicole

VERSION 1 – REVIEW

REVIEWER	Todd, Jemma The University of Sydney, School of Psychology
REVIEW RETURNED	02-Sep-2022

GENERAL COMMENTS	This is a well-designed study examining the role of mental defeat and chronic pain over time. The protocol is clear and well-written, and it will make for a rigorous study. A few queries/comments: - Will participants who have suffered a traumatic brain injury be excluded?- I don't believe the question purported to measure attention to pain actually does so. There is little evidence that individuals have good insight into such cognitive processes, so at best this is a measure of self-report attention to pain. Further, the question asks about time thinking about pain, rather than attending to pain, so this could instead be measuring pain-related worry/rumination?- There are certainly benefits to a multi-pronged recruitment approach. I do wonder whether treatment-seeking patients might vary substantially in their pain experience to those recruited through social media, who may have less pain interference and distress. This may be simply something to acknowledge in the final manuscript, although the authors may want to consider whether accounting for recruitment site or nature of pain is still possible at this stage of the project.- I am not sure whether 'mental defeat' is the most apt construct name for the single item in the diary "how much has the pain brought back to life memories of times when you felt the pain had taken over" – seems more to do with memory bias or imagery to me? I also couldn't locate this item in the pain self-perception questionnaire. Perhaps some more rationale for this item choice could be helpful?- In terms of analysis (p20 lines 17-20), there seems to be a mismatch between exploring "mechanisms of outcomes" and using these variables as DVs. Would a mediation perhaps be more appropriate? How would the proposed analysis provide information about mechanisms of outcomes?- A typo in the diary questions, not sure if these can be amended given the study has already commenced, "since waking today, how physically active has (have) you been"
---

REVIEWER	Gijon-Nogueron, Gabriel University of Malaga, Nursing and Podiatry
REVIEW RETURNED	04-Dec-2022

GENERAL COMMENTS	It has been a pleasure to review your paper about “Investigating Mental Defeat in Patients with Chronic Pain: Protocol for a Longitudinal Experience Sampling Study.” I only see a few points that should be changed to be accept it. Introduction: Can you include a sentence at the end of introduction why is important your study and what add new? Method: In general, the different sections are mixing, can you clarify all the information in their sections? It will be easier to read and understand  - 2.1 Study design: the text is what you will do but not the type of design, please can you change it? - Exclusion criteria: If you said that the pain no cancer, why is an exclusion criteria life-threatening conditions (e.g., cancer)? - 20% of attrition, do you think that is it very high? - Page 9 line 20-37, I want to understand that it’s the section of sample size but it is not clear. Can you clarify the text? New section  - Can you include limitations section?
--

VERSION 1 – AUTHOR RESPONSE

Reviewer 1

Dr. Jemma Todd, The University of Sydney

This is a well-designed study examining the role of mental defeat and chronic pain over time. The protocol is clear and well-written, and it will make for a rigorous study.
Thank you for the kind comment.

A few queries/comments:

- Will participants who have suffered a traumatic brain injury be excluded?

Our screening procedure is carried out by a member of the research team based on self-report. Traumatic brain injury is a significant trigger of pain as well as a long-term comorbid condition. If a potential participant discloses they have sustained/have history of a traumatic brain injury, they will be excluded under the first exclusion criteria in Table 1: *“Have any significant comorbid psychiatric (e.g., psychosis), medical (e.g., coronary heart diseases), neurological (e.g., Alzheimer’s, Parkinson’s, Epilepsy) or life-threatening conditions that would impact pain experience, impede the ability to provide informed consent or complete the study.”*

To improve clarity, we have now specified as a note under Table 1 that the examples in the criteria are not exhaustive, and that eligibility is determined on a case-by-case basis by the research team (see lines: 168-169).

- I don’t believe the question purported to measure attention to pain actually does so. There is little evidence that individuals have good insight into such cognitive processes, so at best this is a measure of self-report attention to pain. Further, the question asks about time thinking about pain, rather than

attending to pain, so this could instead be measuring pain-related worry/rumination? We have amended the manuscript to make it clear that this question is a measure of self-reported attention to pain as well as highlighting the self-report nature of the daily survey in general (see lines: 226, 252). We will be sure to also highlight this in subsequent empirical papers and our analyses.

The attention to pain question is deliberately worded in a way to tap into the amount of time spent on thinking about pain, regardless of the tone/nature of the thoughts. This we believe sets it apart from pain-related worry or rumination, because worry is defined as a chain of thoughts and/or imagery concerning possibility of negative or dangerous events in the future (Borkovec, Ray, & Stober, 1998; Borkovec, Robinson, Pruzinsky, & DePree, 1983), whereas rumination is defined as a repetitive cognitive process with a focus on past mistakes and failures (Nolen-Hoeksema et al., 2008). At the core of these cognitive processes are the negative tone and repetitive nature of the thought, that the attention to pain question does not measure. Interestingly, measures specifically designed to measure worry (e.g. the Penn State Worry Questionnaire; the Brief Measure of Worry Severity) and rumination (e.g. Rumination Response Scale) do not include any items relating to “thinking about pain”. Based on this, we feel *“how much time have you been thinking about your pain?”* is a justifiable momentary assessment that assesses the level of attention to pain. Further research would be needed to clarify to what extent people have good enough insights for reporting their awareness, and to do so reliably. Data from the current study may help to examine the latter point.

- There are certainly benefits to a multi-pronged recruitment approach. I do wonder whether treatment-seeking patients might vary substantially in their pain experience to those recruited through social media, who may have less pain interference and distress. This may be simply something to acknowledge in the final manuscript, although the authors may want to consider whether accounting for recruitment site or nature of pain is still possible at this stage of the project.

Thank you for highlighting this. We agree there could potentially be a difference in treatment-seeking vs non-treatment-seeking individuals, which we will acknowledge in subsequent manuscripts. We have data to distinguish participants' origins in terms of how they were recruited (e.g. social media, recruitment sites, in-person pain clinics etc) but we do not explicitly ask people if they are treatment-seeking vs non-treatment-seeking. We also have measures in the longitudinal questionnaire/screening survey that can be used as indicators of whether a participant is on a stable treatment regime and will run sensitivity analyses to determine any covariance that may need to be accounted for in our main analyses.

- I am not sure whether ‘mental defeat’ is the most apt construct name for the single item in the diary “how much has the pain brought back to life memories of times when you felt the pain had taken over” – seems more to do with memory bias or imagery to me? I also couldn't locate this item in the pain self-perception questionnaire. Perhaps some more rationale for this item choice could be helpful?

Thanks for the comment. To ensure clarity in our writing and to address your previous point, we have specified in the manuscript that the daily surveys are self-report measures and that we have included proxy-measures for some variables (lines: 226-229).

The mental defeat proxy-measure *“how much has the pain brought back to life memories of times when you felt the pain had taken over?”* is worded as such to avoid causing unnecessary distress to participants through repeat exposure – as this question is presented as part of the 3x daily-surveys throughout the study (total of 21 times per tracking period). Applying the full PSPS would not be feasible within an ESM design, as the scale is 24-items long (Tang, 2007). Additionally we felt taking one single item from the PSPS would not fully capture the same meaning and qualitative properties of the concept of mental defeat for non-academic participants (Tang et al., 2009). As a result, we developed this proxy measure as it still conveys the qualitative meaning behind the concept of mental defeat through “pain taking over”. The item was discussed comprehensively by members of the

research team and accepted by PPI representatives as part of the piloting process for the present study, which we have now outlined in the manuscript as well (lines: 231-233).

Furthermore, we would like to note that as part of the wider Warwick Study of Mental Defeat in Chronic Pain Project, we ask participants to complete the full PSPS as part of the study which maps to within +/- 8-weeks of their ESM participation which means we have the potential to run analyses to establish a correlation of our proxy mental defeat measure and the full PSPS score.

- In terms of analysis (p20 lines 17-20), there seems to be a mismatch between exploring “mechanisms of outcomes” and using these variables as DVs. Would a mediation perhaps be more appropriate? How would the proposed analysis provide information about mechanisms of outcomes? Thanks for the opportunity to clarify this. The proposed analyses are an approach that would allow us to first look at inter-association and interactions between a large number of variables, simultaneously. Linear mixed models with a time-lagged design will be fit to the data, and estimate the relationships between all variables directly (Epskamp, 2020). The ml-var/gaussian graphical models (GGM) will provide an exploratory approach in the first instance, by studying the network of temporal and contemporaneous associations between variables. Where appropriate, we will conduct further mediation analyses to more closely examine any significant associations of interest.

- A typo in the diary questions, not sure if these can be amended given the study has already commenced, “since waking today, how physically active has (have) you been”
Thank you for spotting this! We have amended this typo in the manuscript (Table 3, pg 12).

Reviewer 2

Dr. Gabriel Gijon-Nogueron, University of Malaga

Comments to the Author:

Dear authors:

It has been a pleasure to review your paper about “Investigating Mental Defeat in Patients with Chronic Pain: Protocol for a Longitudinal Experience Sampling Study.” I only see a few points that should be changed to be accept it.

Thank you for the kind comment.

Introduction: Can you include a sentence at the end of introduction why is important your study and what add new?

We have adjusted a pre-existing sentence to more explicitly highlight the novelty of this study near the end of the introduction: *“This study will [be the first to utilise] experience sampling methodology (ESM) involving in vivo data gathering (using actigraphy, sleep diaries and daily online surveys) to examine the day-to-day association between mental defeat, symptoms, distress, and disability associated with chronic pain”* (lines: 122-126).

Method: In general, the different sections are mixing, can you clarify all the information in their sections? It will be easier to read and understand

Regarding the methods, we have addressed your following 5 points individually (see below comments).

- 2.1 Study design: the text is what you will do but not the type of design, please can you change it?

We have added the following sentence to the design section to specify the approach: *“This study utilises a within-study design that uses an experience sampling methods (ESM) approach* (Larson &

Csikszentmihalyi, 2014)” (lines: 147-148).

- Exclusion criteria: If you said that the pain no cancer, why is an exclusion criteria life-threatening conditions (e.g., cancer)?

Thanks for spotting the repetition, we have now removed the example “cancer” from the exclusion criteria section of Table 1 to avoid duplication (pg. 7 & 8).

- 20% of attrition, do you think that is it very high?

Based on previous experience sampling studies conducted by the PI (see: Tang et al., 2012; Tang & Sanborn, 2014) as well as a recent meta-analysis (Wrzus & Neubauer, 2022) where similar rates of attrition occurred, we have accounted for 20% attrition in our recruitment target. Previous studies that are longitudinal in nature often report attrition of 30-70% (Gustavson et al., 2012). As the present study uses a prospective ESM framework it is particularly labour-intensive for participants and as such, we expect attrition due to loss-to-follow-up (e.g. not completing the second time-point of experience sampling) in addition to reduced mean compliance (e.g. skipping/not fully completing daily survey(s) and/or sleep diaries). Previous studies investigating attrition in ESM studies, have shown that achieving 100% compliance is unlikely, average compliance and retention rates are 79.7% (Vachon et al., 2019). Based on the available evidence and given the nature our study, we anticipate a ~20% attrition rate. The proposed sample size (accounting for 20% attrition) has also been independently approved by the Medical Research Council, who funded the project this study is part of.

- Page 9 line 20-37, I want to understand that it's the section of sample size but it is not clear. Can you clarify the text?

We have amended the title of section 2.2. from “*Participants*” to “*Participants [& Sample Size]*” to improve clarity and highlight this section on sample size (see line: 161). We have also added a new sentence at the beginning of the relevant paragraph (see lines: 171-172) to make it clearer this section is discussing the number of participants we are aiming to recruit to the study and corresponding number of observations (see lines: 171-179 for full section). Regarding the sample size and power for the GGM analyses, we have also specified further details in lines 413-415.

New section

- Can you include limitations section?

Yes of course. We previously incorporated a limitations section into the standard “Strengths and Limitations” bullet points which appear under the abstract (lines: 56-69) as this is in line with BMJ Open’s journal specifications. We have now added a limitation section to the manuscript (see new “Section 5” – lines 435-450).

References:

- Epskamp, S. (2020). Psychometric network models from time-series and panel data. *Psychometrika*, 85(1), 206–231. <https://doi.org/10.1007/s11336-020-09697-3>
- Gustavson, K., Von Soest, T., Karevold, E., & Roysamb, E. (2012). Attrition and generalizability in longitudinal studies: Findings from a 15-year population-based study and a Monte Carlo simulation study. *BMC Public Health*, 12(1), 1–11. <https://doi.org/10.1186/1471-2458-12-918>
- Larson, R., & Csikszentmihalyi, M. (2014). The experience sampling method. In *Flow and the Foundations of Positive Psychology: The Collected Works of Mihaly Csikszentmihalyi* (Vol. 15,

pp. 21–34). Springer Netherlands. https://doi.org/10.1007/978-94-017-9088-8_2

Tang, N. K. Y., Goodchild, C. E., Sanborn, A. N., Howard, J., & Salkovskis, P. M. (2012). Deciphering the temporal link between pain and sleep in a heterogeneous chronic pain patient sample: a multilevel daily process study. *Sleep*, 35(5), 675–687. <https://doi.org/10.5665/SLEEP.1830>

Tang, N. K. Y., Salkovskis, P. M., Hodges, A., Soong, E., Hanna, M. H., & Hester, J. (2009). Chronic pain syndrome associated with health anxiety: A qualitative thematic comparison between pain patients with high and low health anxiety. *British Journal of Clinical Psychology*, 48(1), 1–20. <https://doi.org/10.1348/014466508X336167>

Tang, N. K. Y., & Sanborn, A. N. (2014). Better Quality Sleep Promotes Daytime Physical Activity in Patients with Chronic Pain? A Multilevel Analysis of the Within-Person Relationship. *PLOS ONE*, 9(3), e92158. <https://doi.org/10.1371/JOURNAL.PONE.0092158>

Vachon, H., Viechtbauer, W., Rintala, A., & Myin-Germeys, I. (2019). Compliance and retention with the experience sampling method over the continuum of severe mental disorders: Meta-analysis and recommendations. *Journal of Medical Internet Research*, 21(12). <https://doi.org/10.2196/14475>

Wrzus, C., & Neubauer, A. B. (2022). Ecological Momentary Assessment: A Meta-Analysis on Designs, Samples, and Compliance Across Research Fields. *Assessment*. https://doi.org/10.1177/10731911211067538/ASSET/IMAGES/LARGE/10.1177_10731911211067538-FIG6.JPEG

VERSION 2 – REVIEW

REVIEWER	Gijon-Nogueron, Gabriel University of Malaga, Nursing and Podiatry
REVIEW RETURNED	30-Dec-2022
GENERAL COMMENTS	Dear Author, thank you for do all the suggestions and changes that I recommended. The paper can accepted in this form